# Antidiarrheal and Cardio-Depressant Effects of *Himalaiella heteromalla* (D.Don) Raab-Straube: In Vitro, In Vivo, and In Silico Studies

**DOI:** 10.3390/plants11010078

**Published:** 2021-12-27

**Authors:** Fatima Saqib, Faisal Usman, Shehneela Malik, Naheed Bano, Najm Ur-Rahman, Muhammad Riaz, Romina Alina Marc (Vlaic), Crina Carmen Mureşan

**Affiliations:** 1Faculty of Pharmacy, Bahauddin Zakariya University, Multan 60000, Pakistan; faisal.usman@bzu.edu.pk (F.U.); shehneelamalik1@gmail.com (S.M.); 2Faculty of Veterinary and Animal Sciences, MNS-University of Agriculture, Multan 60000, Pakistan; bnaheed61@gmail.com; 3Department of Pharmacy, Shaheed Benazir Bhutto University, Sheringal 18050, Pakistan; najm@sbbu.edu.pk (N.U.-R.); pharmariaz@gmail.com (M.R.); 4Food Engineering Department, Faculty of Food Science and Technology, University of Agricultural Sciences and Veterinary Medicine, 400372 Cluj-Napoca, Romania; crina.muresan@usamvcluj.ro

**Keywords:** antidiarrheal, calcium ion channel, cardio-depressant, *Himalaiella heteromalla*

## Abstract

*Himalaiella heteromalla* (D.Don) Raab-Straube is a commonly used remedy against various diseases. Crude extract and fractions of *H. heteromalla* were investigated for a gastrointestinal, bronchodilator, cardiovascular, and anti-inflammatory activities. *H. heteromalla* crude extract (Hh.Cr) relaxed spontaneous contractions and K^+^ (80 mM)-induced contraction in jejunum tissue dose-dependently. The relaxation of K^+^ (80 mM) indicates the presence of Ca^++^ channel blocking (CCB) effect, which was further confirmed by constructing calcium response curves (CRCs) as they caused rightward parallel shift of CRCs in a manner comparable to verapamil, so the spasmolytic effect of Hh.Cr was due to its CCB activity. Application of Hh.Cr on CCh (1 µM) and K^+^ (80 mM)-induced contraction in tracheal preparation resulted in complete relaxation, showing its bronchodilator effect mediated through Ca^++^ channels and cholinergic antagonist activity. Application of Hh.Cr on aortic preparations exhibited vasorelaxant activity through angiotensin and α-adrenergic receptors blockage. It also showed the cardio suppressant effect with negative chronotropic and inotropic response in paired atrium preparation. Similar effects were observed in in vivo models, i.e., decreased propulsive movement, wet feces, and inhibition of edema formation.

## 1. Introduction

*Himalaiella heteromalla* (D.Don) Raab-Straube (*Asteraceae*), commonly known as Batula, is found in low-temperature regions of Asia, Europe, and North America [1]. *Himalaiella heteromalla* is a rich source of chlorojanerin, arctigenin [2] glycosides, alkaloids, terpenoids, saponins, flavonoids, sesquiterpene lactones, and arctiin [3]. Gao et al. [4] and Kang et al. [5] reported arctigenin and its glycoside, arctiin, have anti-inflammatory activities by inhibiting iNOS and exerting vasodilation effect, while Hayashi et al. [6] reported the anti-viral activity against influenza A virus.

Traditionally, *H. heteromalla* is used in herbal products to treat fever, menstruation, circulation, pain, and rheumatic arthritis [7]. It is used in wounds, cuts, and fever [8]. The leaf paste and mustard oil mixture are used for wounds and leukoderma. It has carminative property, used for coeliac diseases [9,10]. It is used to remedy burning parts of the body, menstrual problems, piles, psoriasis, rheumatoid arthritis, cardiotonic cough with cold, and altitude sickness, and provide anticancer and anti-fatigue actions [11]. It is used as an anti-inflammatory and prevents ischemic stroke [3]. Therefore, *H. heteromalla* was investigated in in vitro, in vivo, and in silico models as a possible tool to treat gastrointestinal, cardiovascular, respiratory, and inflammatory ailments.

## 2. Results

### 2.1. Phytochemical Analysis of Himalaiella heteromalla

The preliminary phytochemical analysis of Hh.Cr confirmed the presence of glycosides, saponins, alkaloids, and flavonoids.

### 2.2. HPLC Separation of Phenolic Acids and Flavonoids

The separation factor and resolution of all separated compounds were >1.0 and >1.5, respectively. The reproducibility of separate components was also good with RSD < 2% (run to run) and 2.7% (day to day) [12]. The HPLC chromatograms for the identified phenolic and flavonoid compounds are shown in Figure 1, and Table 1 presents the phenolic and flavonoid compounds identified in Hh.Cr. The most abundant phenolic compounds were gallic acid (184.98 µg/g), hydroxybenzoic acid (6.8 µg/g), and vanillic acid (8.1 µg/g); while the identified flavonoid compound was catechin (160.37 µg/g).

### 2.3. Effect on Jejunum Preparations

The Hh.Cr and its ethyl acetate (Hh.Ea) fraction showed the relaxant effect on exposure to the rhythmic contraction of jejunum preparations in organ bath within concentration range 0.01 to 0.3 mg/mL with EC_50_ 0.06 mg/mL (95% CI: 0.045–0.080 mg/mL; n = 5) and 0.01 to 0.1 mg/mL with EC_50_ 0.032 mg/mL (95% CI: 0.021–0.51 mg/mL; n = 5), respectively similar to verapamil with EC_50_ 0.42 µM (95% CI: 0.22–1.27), whereas aqueous fraction *H. heteromalla* failed to complete relaxation of spontaneous contractions of jejunum preparations. Hh.Cr and Hh.Ea also caused a complete relaxation of K^+^ (80 mM) induced spastic contractions at 1 mg/mL with EC_50_ 0.13 mg/mL (95% CI: 0.088–0.220 mg/mL; n = 5) and 0.3 mg/mL with EC_50_ 0.06 mg/mL (95% CI: 0.045–0.0.089 mg/mL; n = 5), respectively similar to verapamil with EC_50_ 0.251 µM (95% CI: 0.082–0.784). Furthermore, Hh.Cr showed a rightward shift of calcium CRCs which confirm the presence calcium ion channel blockade activity in Hh.Cr, similar to verapamil (Figure 2 and Figure 3).

### 2.4. Effect on Tracheal Preparations

The Hh.Cr and its ethyl acetate (Hh.Ea) fraction showed the relaxant effect on tracheal preparations, when exposed K^+^ (80 mM) and CCh (1 µM)-induced contractions. Hh.Cr and its Hh.Ea fraction relaxed the K^+^ (80 mM) induce contractions at 0.3 mg/mL with EC_50_ 0.19 mg/mL (95% CI: 0.099–0.452; n = 5) and 0.1 mg/mL with EC_50_ 0.042 mg/mL (95% CI: 0.024–0.072 mg/mL; n = 5), respectively. Hh.Cr and its Hh.Ea fraction also relaxed the CCh (1 µM) induce contractions at 1 mg/mL with EC_50_ 0.23 mg/mL (95% CI: 0.158–0.357 mg/mL; n = 5) and 0.3 mg/mL with EC_50_ 0.155 mg/mL (95% CI: 0.076–0.302 mg/mL; n = 5), respectively. Similarly, verapamil also caused relaxation of K^+^ (80 mM) and CCh (1 µM) induced contractions with respective EC_50_ 0.82 µM (95% CI: 0.82–0.82 µM) and EC_50_ values of 2.35 µM (95% CI: 02.32–2.39 µM). The aqueous fraction (Hh.Aq) exerted partially relaxation of K^+^ (80 mM) and CCh (1 µM)-induced contractions on tracheal preparation (Figure 4).

### 2.5. Effect on Aortic Preparations

The crude extract (Hh.Cr) and its ethyl acetate (Hh.Ea) fraction showed the relaxant effect on aortic preparations, when exposed K^+^ (80 mM) and PE (1 µM) induced contractions. Hh.Cr and its Hh.Ea fraction relaxed the K^+^ (80 mM) induce contractions at 3 mg/mL with EC_50_ 2.88 mg/mL (95% CI: 2.106–4.156 mg/mL; n = 5) and 1 mg/mL with EC_50_ 0.148 mg/mL (95% CI: 0.09491–0.2332 mg/mL; n = 5), respectively. Hh.Cr and its Hh.Ea fraction also relaxed the PE (1 µM) induce contractions at 5 mg/mL with EC_50_ 15.53 mg/mL (95% CI: 7.965 to 62.27 mg/mL; n = 5) and 3 mg/mL with EC_50_ 4.2 mg/mL (95% CI: 2.991 to 6.670 mg/mL; n = 5), respectively. Similarly, verapamil also caused relaxation of K^+^(80 mM) and PE (1 µM) induced contractions with respective EC_50_ 1.054 µM (95% CI: 0.45–5.68) and 0.764 µM (95% CI: 0.33–68.8). The aqueous fraction of (Hh.Aq) partially exerted relaxation of K^+^(80 mM) and PE (1 µM) induced contractions on aortic preparation (Figure 5).

### 2.6. Effect on Atria Preparations

The crude extract (Hh.Cr) and its ethyl acetate (Hh.Ea) fraction caused the negative chronotropic effect (i.e., decrease in heart rate) and negative inotropic effect (i.e., force of contraction) on atrium preparation [12]. Hh.Cr and its Hh.Ea fraction showed negative inotropic effect within concentration range 0.01–5.0 mg/mL with EC_50_ 0. 9 mg/mL (95% CI: 0.375–1.356 mg/mL; n = 3) and 0.01–3.0 mg/mL with EC_50_ 0.7 mg/mL (95% CI: 0.265–0.586 mg/mL; n = 3), respectively. Hh.Cr and its Hh.Ea fraction showed the negative chronotropic effect within concentration range 0.01–5.0 mg/mL with the EC_50_ 0.5 mg/mL (95% CI: 0.406–0.680 mg/mL; n = 3) and 0.01–3.0 mg/mL with the EC_50_ value calculated to be 0.4 mg/mL (95% CI: 0.106–0.050 mg/mL; n = 3), respectively. Similarly, verapamil also showed negative inotropic and chronotropic effect with concentration range 0.01–1.0 mg/mL with EC_50_ value of 0.053 µM (95% CI: 0.034–0.084 µM; n = 3) and 0.01–0.3 mg/mL with EC_50_ 0.037 µM (95% CI: 0.024–0.045 µM; n = 3). The aqueous fraction (Hh.Aq) exerted partially negative inotropic and chronotropic effect on atrium preparation with in concentration range 3–10 mg/mL with EC_50_ 1.02 mg/mL (95% CI: 0.485–0.856 mg/mL; n = 3) and 3–10 mg/mL with 1.14 mg/mL (95% CI: 0.575–1.756 mg/mL; n = 3), respectively (Figure 6).

### 2.7. Antiperistalsis Activity

The crude extract (Hh.Cr) showed a significant antiperistalsis response in mice with less distance traveled by charcoal meal as compared to control (33 ± 2.3%). The group was treated with 400 mg/kg of Hh.Cr and CCh (3 mg/kg) and peristaltic movements were significantly decreased by 1.2 ± 0.37 and 8.6 ± 1.8, respectively (Figure 7).

### 2.8. Antidiarrheal Activity

The crude extract (Hh.Cr) showed a significant antidiarrheal response in rats with fewer wet fecal masses than control (15.60 ± 1.4). The group was treated with 400 mg/kg of Hh.Cr and loperamide (3 mg/kg) showed highly significant anti-diarrheal effect 1.2 ± 0.37 and 0.8 ± 0.37, respectively (Figure 7).

### 2.9. Anti-Inflammatory Activity

The crude extract (Hh.Cr) showed a significant anti-inflammatory response in rats with inhibition of edematous volume of hind paw as compared to control (2.93 ± 0.2 mL) at maximum duration. The Hh.Cr inhibited the paw edema at 100 mg/kg as 0.96 ± 0.08 mL, 1.10 ± 0.01 mL, 1.16 ± 0.01 mL, 1.12 ± 0.015 mL, and 1.23± 0.01 mL at 0, 1, 2, 3, and 4 h duration, respectively, whereas at dose 300 mg/kg, it showed maximum inhibition, i.e., 0.87 ± 0.01 mL, 0.86 ± 0.01 mL, 0.86 ± 0.03 mL, 0.91 ± 0.01 mL, 1.04 ± 0.001 mL, and 1.15 ± 0.02 mL at 0, 1, 2, 3, and 4 h duration, respectively. Hh.Cr inhibited the edematous volume as similar to the aspirin, i.e., 1.05 ± 0.02 mL (Figure 7).

### 2.10. In Silico Studies

The docking calculations are beneficial to predict ligand pose within the binding site of the target protein. The involvement of physical energies terms (i.e., solvation energy) with suitable force field make docking calculation of compounds more acceptable with accuracy (Table 2) [12,13,14].

*Molecular docking for Muscarinic M3 receptor:* The selected compounds were studied against muscarinic M3 (MM3, PDB ID: 4U14)) for antispasmodic activity (Table 2, Figure 8). Arctiin was predicted with the lowest binding energy (ΔG_bind_: −60.79 kcal/mol) with hydrophobic energies ΔG_vdW_ (−55.32 kcal/mol) and ΔG_Lipo_ (−42.47 kcal/mol) major contributors to the ligand binding energy. It formed the two π-donor hydrogen interaction with residue Trp525 and hydrophobic interactions (π-π Stacked Bond: Trp503; π-π T shaped Bond: Tyr148) within the pocket of MM3. Besides these, it also formed the π-Sulfur interaction with residue Cys532. Arctigenin second to arctiin also found potent with have ligand binding energy (ΔG_bind_: −46.56 kcal/mol) mainly contributed with hydrogen bond interaction (∆G_Hbond_: −0.68 kcal/mol) and hydrophobic interaction (ΔG_vdW_: −34.45 kcal/mol and ΔG_Lipo_: −30.61 kcal/mol). It formed hydrophobic π-π T-shaped interaction with residue Trp503 and Trp525 within the protein cleft. Moreover, catechin has the lowest binding energy (ΔG_bind_: 52.22 kcal/mol) with ΔG_vdW_ (−34.12 kcal/mol) and ΔG_Lipo_ (−13.93 kcal/mol) and formed π-π T shaped interaction with residue Tyr506. The ranking orders of ligands with COX-2 is given below: arctiin > arctigenin > catechin > chlorojanerin > cynaropicrin.

*Molecular docking for cyclooxygenase-2 enzyme*: The selected compounds were studied against cyclooxygenase-2 enzyme (COX-2, PDB ID:5IKQ) for anti-inflammatory activity (Table 2, Figure 8). Arctiin was predicted with the lowest binding energy (ΔG_bind_: −41.01 kcal/mol) among the selected compounds. As mentioned earlier, Van der Waals (ΔG_vdW_) and lipophilic interactions (ΔG_Lipo_) are significant contributors to the ligand binding energy. It was observed that the binding energies value of ΔG_vdW_ was −36.96 kcal/mol, and ΔG_Lipo_ was −23.77 kcal/mol, whereas hydrogen bond (∆G_Hbond_) energy contribution was −2.56 kcal/mol. It also formed hydrophobic interactions π-π T shaped interaction with Tyr11. Arctigenin second to arctiin in docking score was found with potential hydrophobic interactions within hydrophobic clefts of COX-2. The ligand binding energy of arctigenin (ΔG_bind_: −27.91 kcal/mol) was driven mainly by these hydrophobic interaction energies; Δ G_vdW_ (−35.44 kcal/mol) and ΔG_Lipo_ (−31.09 kcal/mol) and formed hydrophobic π–σ interaction with Val117. The ranking orders of ligands with COX-2 are given below: arctiin > arctigenin > cynaropicrin > catechin.

*Molecular docking for lipoxygenase-5 enzyme:* The selected compounds were studied against lipoxygenase-2 enzyme (LOX-5, PDB ID: 6N2W) for anti-inflammatory activity (Table 2, Figure 8). The contribution of hydrophobic interactions in ligand binding energy was more abundant within pockets of LOX-5. Arctigenin has higher ligand binding energy (ΔG_bind_: −42.94 kcal/mol) but ranks third in the docking score. The ligand binding energy contributed with hydrogen bond interaction (∆G_Hbond_: −3.06 kcal/mol) and hydrophobic interaction (ΔG_vdW_: −29.61 kcal/mol and ΔG_Lipo_: −18.99 kcal/mol). It also formed hydrophobic interactions (π-π Stacked Bond: His372) within the pocket of LOX-5. Arctiin ranked at first in position docking score with ligand binding energy (ΔG_bind_: −30.76 kcal/mol) which mainly contributed from hydrophobic interaction energies ΔG_vdW_ (−46.53 kcal/mol) and ΔG_Lipo_ (−16.17 kcal/mol), whereas hydrogen bond (∆G_Hbond_) energy contribution was −2.11 kcal/mol. Besides hydrophobic interaction, arctigenin and arctiin also formed electrostatic charge interaction with residue Arg596 and Ile673 within the cleft of COX-2, respectively. Catechin have ligand binding energy (ΔG_bind_: −30.81 kcal/mol) mainly contributed with hydrogen bond interaction (∆G_Hbond_: −2.38 kcal/mol) and hydrophobic interaction (ΔG_vdW_: −32.68 kcal/mol and ΔG_Lipo_: −15.10 kcal/mol). Catechin formed the π–donor hydrogen interaction with residue His372 and hydrophobic interactions (π-π Stacked Bond: His367; π-π T shaped Bond: His372, Trp599) within the pocket of LOX-5. The ranking order of ligands with COX-2 is given below: arctiin > catechin > arctigenin > cynaropicrin > chlorojanerin

## 3. Discussion

*Himalaiella heteromalla* has a potential pharmacological role in the management of various diseases. This research was employed to investigate its pharmacological characteristics. The presence of alkaloids, glycosides, triterpenoids, flavonoids, saponins, sesquiterpene, which play a vital role in the pharmacological potential of *Himalaiella heteromalla*. The HPLC results indicate the presence of gallic acid, catechin, HB acid, and vanillin acid. Gallic acid (3,4,5trihydroxybenzoic acid), a natural polyphenol product, has anti-oxidant, anti-inflammatory, antimicrobial, and radical scavenging activities. Gallic acid is used as a spasmolytic effect on smooth muscle isolated jejunum tissues and trachea by calcium channel blocking activity [15]. Gallic acid is used as an antispasmodic in diarrhea [16]. Gallic acid possesses an anti-inflammatory effect [17]. Catechin abundant flavonoid present in plants, it reported for several gastrointestinal, respiratory, and inflammatory disorders [18,19,20]. Vanilla acid and HB acid are polyphenolics used for gastrointestinal, respiratory, and cardiovascular disorders by spasmolytic effects on isolated tissues of the jejunum, trachea, and aorta [20]. The water content in *H. heteromalla* play a vital role in the biological activities, so it is more important to measure the water content in *H. heteromalla* therefore, infra red radiation can be used to measure water content determination [21].

*Himalaiella heteromalla* crude extract (Hh.Cr) was studied on isolated jejunum preparations to elaborate the mechanism of *H. heteromalla* in gastrointestinal diseases. It is reported that jejunum preparations have rhythmic contractions due to the influx of calcium ions and potassium ions through their respective ions channels. *H. heteromalla* crude extract and its fraction ethyl acetate exerted spasmolytic response in dose concentration when exposed to spontaneous contraction of jejunum preparations [22]. Thus, *H. heteromalla* crude extract and its fraction ethyl acetate showed the antispasmodic response by suppressing rhythmic contractions in jejunum preparations. These results indicate that *H. heteromalla* crude extract and its fraction ethyl acetate decrease or blockade the cytoplasmic free Ca^++^ ions through the blockade of voltage-dependent calcium ion channels. As a result, activation of calmodulin and other contractile proteins, i.e., actin and myosin, does not occur [23]. *H. heteromalla* crude extract and its fraction ethyl acetate and aqueous were exposed to K^+^ (80 mM)-induced contractions on jejunum preparations, *H. heteromalla* crude extract and its fraction ethyl acetate relaxed the K^+^ (80 mM)-induced contractions in dose concentration manner in a tissue organ bath. It was previously reported that K^+^ (80 mM) induces contractions to cause cell depolarization by the influx of calcium ions into the cell through the voltage-gated L-type calcium ion channel [24]. Similar to verapamil, any substance inhibited K^+^ (80 mM)-induced contractions were considered calcium channel blockers (CCB). Thus, *H. heteromalla* crude extract and its fraction ethyl acetate blockade the calcium influx into the cell by alternating or binding with voltage-dependent calcium channels. Furthermore, calcium concentration–response curves (CRCs) were constructed on pretreated Hh.Cr jejunum preparations to confirm the calcium channel blockade activity of Hh.Cr in a tissue organ bath. The results showed that partial blockade with the rightward parallels dose–response curves at low doses while completely blocking the dose–response curves at 0.3 mg/m. Thus, *Himalaiella heteromalla* exhibited a strong calcium antagonistic effect [25].

To evaluate another possible mechanism of *H. heteromalla* crude extract on the gastrointestinal tract, Hh.Cr was studied in antiperistalsis and antidiarrheal in vivo models. *H. heteromalla* crude extract showed the antispasmodic response by inhibiting the traveling of charcoal meal in antiperistalsis activity. *H. heteromalla* crude extract also inhibited diarrheal response in castor oil-induced diarrhea. It decreased the wet feces by inhibiting the electrolyte and water imbalance that may cause diarrhea in rats [26].

*H. heteromalla* crude extract and its fractions ethyl acetate and aqueous were tested for possible bronchodilator activity against CCh (1 μM) and K^+^ (80 mM)-induced contractions on tracheal preparations. The results showed that *H. heteromalla* crude extract and its fractions ethyl acetate exhibited relaxant response against CCh (1 μM) and K^+^ (80 mM)- induced contractions, but a partial relaxant effect was observed by the aqueous fraction. However, EC_50_ of *H. heteromalla* crude extract and its fractions ethyl acetate against K^+^ (80 mM)-induced contractions that were more minor than CCh-induced contractions, similar to that of verapamil. CCh is a cholinergic agonist which causes smooth muscle contraction through activation of muscarinic receptors. Hence, *H. heteromalla* crude extract and its fractions ethyl acetate showed bronchodilator response was found due to Ca^++^ ion channel and muscarinic receptor blockade. Nowadays, Ca^++^ channel blockers and muscarinic antagonists are used to treat the relief from respiratory diseases such as asthma [27,28].

*H. heteromalla* crude extract and its fractions ethyl acetate and aqueous were tested for possible vasorelaxant activity against PE (1 µM) and K^+^ (80 mM)-induced contractions on aortic preparations. The results showed that *H. heteromalla* crude extract and its fractions ethyl acetate exhibited a relaxant response against PE (1 µM) and K^+^ (80 mM)-induced contractions, similar to verapamil. *H. heteromalla* aqueous fraction partially relaxed the PE and K^+^ (80 mM) induced contractions. Relaxation of the PE (PE) and K^+^ (80 mM)-induced contractions indicates a blockade of intracellular Ca^++^ influx by blocking Ca^++^ channels. Ca^++^ channel blockers are essential drugs used clinically to manage angina and hypertension [29,30].

*H. heteromalla* crude extract and its fractions ethyl acetate and aqueous were tested on paired atrium for possible effects on force and rate of atrial contractions. *H. heteromalla* crude extract and its fractions ethyl acetate and aqueous showed cardio suppressant response via blocking calcium channels, hence Hh.Cr and its fractions were found with adverse inotropic and chronotropic effects on the paired atrium [31].

*Himalaiella heteromalla* crude extract was tested for anti-inflammatory activity. It was found that Hh.Cr blocked the release of inflammatory mediators in rat paw edema and other models. It is reported that carrageenan acetic acid and formalin release inflammatory mediators such as bradykinin, histamine, TNF, IL-1b, IL-6, PEG2, and TNF were blocked by crude extract of crude extract *Himalaiella heteromalla.* The reduction in inflammatory mediators by carrageenan, inducing the rat’s paw edema model to show that Hh.Cr inhibits factors that cause inflammation and swelling. On the other side, pain sensation is a significant indicator in the inflammation process, which Hh blocked Cr, so that it exhibited analgesic activity. This anti-inflammatory result was compared with standard drug analgesic aspirin, reducing all models’ inflammation and pain sensations. The comparative results in between aspirin and Hh.Cr showed that Hh.Cr exhibited thde same potential as aspirin to reduce the pain and inflammation via blockading inflammatory mediators [32].

Molecular docking is a helpful tool to predict the possible mechanism of actions of the selected compounds of various pharmacological studies—the present study correlated and defined antispasmodic and anti-inflammatory activities of *Himalaiella heteromalla.* The five compounds of *H. heteromalla* were studied for cyclooxygenase 2, lipoxygenase 5, and muscarinic M3 receptor. The docking calculations of these compounds indicate the presence of antispasmodic and anti-inflammatory activities, which were previously proven in experimental studies. Arctiin and arctigenin were more potent compounds responsible for these activities. These results conclude that these compounds interact with cyclooxygenase 2, lipoxygenase 5, and muscarinic M3 receptor to exert the activity. As mentioned earlier, Gao et al. [4] reported that arctigenin and arctiin have anti-inflammatory and vasodilation properties and help treat acute lung injury, local edema, brain trauma, and colitis. These studies support the potent results of arctigenin and arctiin in silico studies against major inflammatory proteins COX-2 and LOX-5.

## 4. Materials and Methods

### 4.1. Extract Preparation

*Himalaiella heteromalla* (D.Don) Raab-Straube was collected from hilly areas of Islamabad and identified by Dr. Zafarullah Zafar, taxonomist, Institute of Pure and Applied Biology, and submitted with voucher no: http://www.theplantlist.org/tpl1.1/record/gcc-138921 dated 18 June 2018. Plant material was ground through a herbal grinder for coarse powder, then powder (1 kg) was macerated in methanol aqueous (70:30) for maceration in an amber color glass bottle for three days at room temperature and periodically shaken 3–4 times a day. The solvent was filtered to remove plant debris with muslin cloth and Whattman-1 filter paper. This procedure was replicated thrice, and the filtrate obtained by all steps was combined and processed in a rotary evaporator (BUCHI) under reduced pressure at 36 ± 2 °C to obtain a brownish colored semi-solid (Hh.Cr) and stored at −20 °C in an airtight jar with a percentage yield of 12%. The Hh.Cr (20 g) was subjected to solvent–solvent extraction with ethyl acetate and distilled water to produce an ethyl-acetate fraction (Hh.Ea) and aqueous fraction (Hh.Aq) with approximately 5.5% and 40% yield, respectively. *H. heteromalla* crude extract (Hh.Cr) and its fractions were moderately soluble in aqueous. All dilutions were prepared fresh on the day of the experiment.

### 4.2. Animal Housing

Both sexes of albino mice (weight: 20–30 g), rats (weight: 150–200 g), and rabbits (weight: 1–1.8 kg) were used in this study and kept under controlled housing conditions with a temperature of 23 ± 3 °C in the animal house of the Faculty of Pharmacy, Bahauddin Zakariya University, Multan. Before the experiment, animals were deprived of food overnight but had free access to water. For in vitro experimentation, rabbits were sacrificed following a blow, while mice and rats were killed by cervical dislocation. All the experimentations were performed under rules specified by the Institute of Laboratory Animal Resources, Commission on Life Sciences (NRC, 1996) endorsed by the Ethical Committee of Bahauddin Zakariya University, Multan.

### 4.3. Chemicals

All the chemicals used in this study have high purity with research-grade quality. Acetylcholine (Ach), aspirin, carbamylcholine chloride HCl, Carbachol (CCh), verapamil HCl, phenylephrine (PE) were purchased from Sigma Chemical Company, St. Louis, MO, USA. While Potassium dihydrogen phosphate, magnesium chloride, sodium bicarbonate, sodium chloride, magnesium sulfate, sodium dihydrogen phosphate, calcium chloride, potassium chloride, ethylenediaminetetraacetic acid (EDTA), glucose were purchased from Merck, Dermstadat Germany. Furthermore, loperamide, and dicyclomine were supplied by Sigma Chemical company, St. Louis, MO, USA).

### 4.4. Qualitative Phytochemical Detection

The qualitative phytochemical investigation of *H. heteromalla* was performed to identify alkaloids, glycosides, anthraquinones, terpenes, saponins, flavonoids, and phenols.

### 4.5. HPLC Separation of Phenolic Acids and Flavonoids

The phenolic acids and flavonoids components in *Himalaiella heteromalla* were quantified by developing a binary gradient solvent system to run in Chromera HPLC system (Perkin Elmer, Houston, TX, USA) consisting of Felexer Binary Liquid chromatography (LC) pump coupled with UV/Vis LC Detector (Shelton, CT, USA) which was operated with the help of a software. HPLC system consisted of a C-18 column (250 × 4.6 mm internal diameter) with a thickness of 5 µM film. The mobile phase consisted of solvent A (methanol (30): acetonitrile (70)) and solvent B (0.5% glacial acetic acid in double-distilled water), mobile phase run at flow rate 0.08 mL/min, and data was recorded at 275 nm of UV spectra. The peaks and retention times of phenolic acids and flavonoids of *H. heteromalla* were matched with external standards to quantify the components [12]. The resolution and separation factor was used to determine HPLC separation efficiency.

### 4.6. In Vitro Experiments

The physiological response of tissues was recorded with isotonic and force-displacement isometric transducers amplified with acquisition system Power Lab (AD Instruments, Bella Vista, NSW, Australia) coupled with a computer having Lab chart Pro. The effect was taken as percent change on the part of test substance recorded instantly preceding a dose of test substance [22].

#### 4.6.1. Isolated Rabbit Jejunum Preparation

The jejunum was dissected from a rabbit; the adhesive fatty tissues were carefully removed, and then ~2 cm long piece of jejunum was prepared. This tissue was hung in an organ bath containing Tyrode’s solution with a continuous supply of carbogen (95% O_2_ + 5% CO_2_) at 37 °C and equilibrated for 30 min. Acetylcholine (1 µM) was added to spontaneous rhythmic contractions of jejunum for control response and washed it. The Hh.Cr was added cumulatively for antispasmodic effect. The spontaneous contractions jejunum preparation was exposed to K^+^ (80 mM) induced contraction for estimation of CCB activity [33].

The extract was exposed to calcium concentration response curves (CRCs) for further confirmation. The jejunum preparation was stabilized in Tyrode’s solution, subsequently replaced with calcium-free Tyrode’s solution with EDTA (0.1 mM) to remove calcium from tissue. Afterward, with an incubation duration of 40 min, two superimposable control calcium CRCs were constructed in an organ bath, then tissue was incubated with the plant exact for one h, and calcium CRCs were obtained and compared to control. The calcium CRCs were recorded in the presence of different concentrations of plant extract.

#### 4.6.2. Isolated Rabbit Tracheal Preparations

The trachea was dissected from a rabbit for bronchodilator activity, and the 2 mm tracheal ring tissue was prepared. A longitudinal incision was made opposite the smooth muscle layer to form a strip. This tracheal preparation was hung in an organ bath containing Krebs’s solution with a continuous supply of carbogen at 37 °C. Preload tension (1 g) was applied and allowed to equilibrate for 60 min prior to the dose of any drug. The tracheal preparation was exposed to CCh (1 µM), and K^+^ (80 mM) induced contraction for bronchodilator activity in a cumulative manner.

#### 4.6.3. Isolated Rabbit Paired Atria Preparations:

The heart was dissected from a rabbit for cardiac activity, and the ventricles were carefully removed to isolate paired atria. This atrium preparation was hung in an organ bath containing Krebs’s solution with a continuous supply of carbogen at 34 °C. Then, 1 g preload tension was applied and allowed to equilibrate for 30 min prior to the dose of any drug. The isolated atrium preparation was exposed for possible cardiac effects in a cumulative fashion, and changes in rate and force of contractions were observed.

#### 4.6.4. Isolated Rabbit Aorta Preparations

For vasorelaxant activity, the thoracic aorta was dissected from a rabbit, carefully removed the adhesive fatty tissues, and prepared 2–3 mm aortic rings. This aortic preparation was hung in an organ bath containing Krebs’s solution with a continuous supply of carbogen. Preload tension (2 g) was applied and allowed to equilibrate for 60 min prior to the dose of any drug. The isolated aortic preparation was exposed to *H. heteromalla* for possible vasorelaxant effects in a cumulative manner. Further to define the possible mechanism, *H. heteromalla* was challenged to PE (1 µM), and K^+^ (80 mM) induced contraction for the possible activity of *H. heteromalla* in a cumulative manner.

### 4.7. In Vivo Activities

#### 4.7.1. Antiperistalsis Activity

Antiperistalsis activity was performed according to the method prescribed by Wahid et al. [12]. Mice (25) of either sex were divided into 5 groups, i.e., control (0.9% normal saline), standard drug (CCh10 mg/kg), and *H. heteromalla* doses (100, 200, and 400 mg/kg). After 15 min of administering the test or standard material orally, each animal received 0.3 mL of the charcoal meal (10% gum acacia, 20 starch, and 10% vegetable charcoal) in distilled water. Thirty minutes later, mice were killed, and the abdomen was incised to excise the whole small intestine. The distance from the pylorus region was measured to the front of the charcoal meal.

#### 4.7.2. Antidiarrheal Activity

The antidiarrheal activity was performed according to the method prescribed by Wahid et al. [12] with modifications. Mice (20) of either sex were divided into five groups, i.e., negative control (0.9% normal saline), standard drug (loperamide 10 mg/kg), and *H. heteromalla* doses (100, 200, and 400 mg/kg). After 30 min of dose administration (p.o.), animals received the castor oil (10 mL/kg p.o) and were observed for six hours in cages with a white paper surface with adsorbent properties. The percent inhibition of wet fecal was calculated.

#### 4.7.3. Carrageenan-Induced Rat’s Hind Paw Edema Method

The anti-inflammatory activity was performed [32,34] on 25 rats of either sex divided into 5 groups, i.e., control (0.9% normal saline) standard drug (aspirin 10 mg/kg), and *H. heteromalla* doses (100, 200, and 400 mg/kg). After 30 min of dose administration (p.o), edema was induced by injecting 1% carrageenan in the right hind paw’s sub-planter region and measuring the edema size at up to 4 h through a plethysmometer. The percentage of edema inhibition was calculated.

### 4.8. In Silico Studies

In silico studies were performed according to the method previously reported by Wahid et al. [12] and Sirous et al. [13].

*Ligand Preparation:* The 2D structures of HPLC quantified phytocompounds were retrieved from PubChem (https://pubchem.ncbi.nlm.nih.gov accessed date 20 March 2019) and treated in the LigPrep module of Maestro (Schrodinger suite 2015) to ionization, minimization, and optimization of ligands. The Epik tool of this module was used to generate the ionization state of ligands at cellular pH (7.4 ± 0.5) and applied the OPLS3e force field through the module for minimization and optimization of ligands that produce the lowest energy conformer of ligands.

*Protein Preparation:* For molecular docking, the highest resolution X-ray structures of proteins were downloaded from The Protein Databank (RCSB PDB) (https://www.rcsb.org accessed date 20 March 2019) and subjected to Protein preparation wizard of Maestro (Schrodinger suite 2015). This module processed the protein by adding hydrogen atoms to protein structure, removal of solvents (water) molecules, assigning bond orders, creating disulfide bonds, filling missing side chains and loops, and generating protonation state using Epik tool of protein structures for ligands at the cellular level pH (7.4 ± 0.5). After processing protein structures, these structures were optimized using PROPKA under pH 7.0, and the OPLS3e force field was utilized to perform restrained minimization for energy minimization and geometry optimization of protein structure.

*Molecular Docking and Receptor grid generation:* The active sites of protein structures for molecular docking were defined in the Receptor Grid Generation module of Maestro (Schrodinger suite 2015). A cubic grid box of each protein was defined with the help of a literature survey and with a selection of previously bonded ligands of proteins. The length of the grid box was adjusted to the length of 16 Å. The potential of nonpolar parts of the receptor was decreased to scaling factor 1.0 Å on Van der Waals radius of nonpolar atoms of protein having partial atomic charge cut-off 0.25 Å.

For molecular docking, the prepared ligands and protein structures were subjected to extra precision (XP) mode of Ligand Docking (Glide) module of Maestro (Schrodinger suite 2015) using pre-generated grid file for receptor. Additionally, 0.80 Å scaling factor was adjusted for Van der Waals radii with a partial charge cut-off of 0.15 Å. The docking results were subjected to the Prime MM-GBSA module to calculate the binding energies of ligands with protein structure using the VSGB solvation model with OPLS3e force field.

*Inhibition Constant (K_i_):* The inhibition constant was determined from the binding free energy of ligand previously generated from Prime MM-GBSA, according to the following equation [12]:∆G = −RT(lnK_i_) or K_i_ = e^(−∆G/RT)^
where ∆G is binding free energy of ligand, R is gas constant (cal·mol^−1^·K^−1^), and T is room temperature (298 Kelvin).

### 4.9. Statistical Analysis

The data were expressed as the mean ± standard error of the mean (S.E.M.) and median effective concentration (EC_50_) with a 95% confidence interval (CI). One-way and two-way ANOVA tests were applied for in vivo experiments. All graphs and data were analyzed with the help of Graph pad prism software (San Diego, CA, USA).

## 5. Conclusions

*Himalaiella heteromalla* exhibited a more spasmolytic effect in ethyl acetate fraction and caused complete relaxation on isolated jejunum, trachea, aorta, and paired atria, supported with in silico studies. *H. heteromalla* proved various disease management-related activities. Further studies could be taken on *Himalaiella heteromalla* for drug discovery for the welfare of human beings.

## Figures and Tables

**Figure 1 plants-11-00078-f001:**
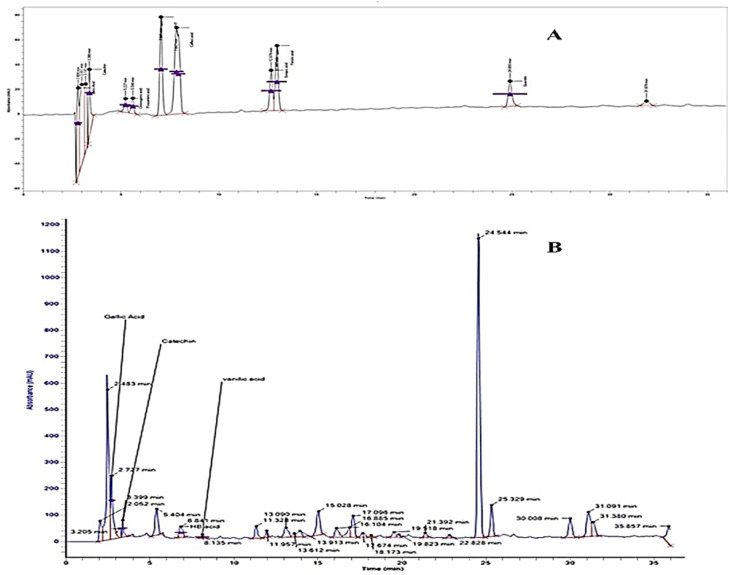
HPLC chromatogram of (**A**) standard phenolic compounds (**B**) Hh.Cr.

**Figure 2 plants-11-00078-f002:**
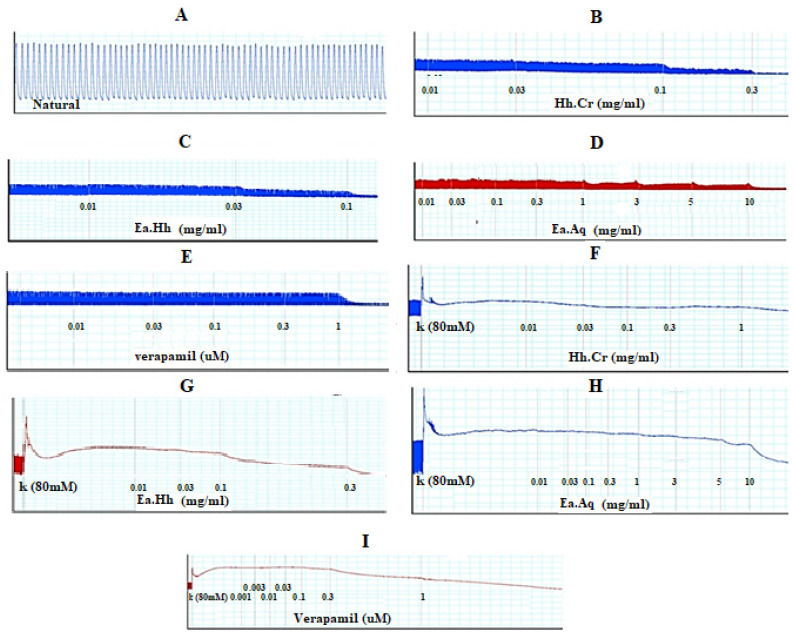
(**A**) Control spontaneous contraction. Effect of (**B**) crude extract (Hh.Cr), (**C**) Ethyl acetate fraction (Ea.Hh), (**D**) Aqueous fraction (Ea.Aq), and (**E**) verapamil on spontaneous. Effect of (**F**) crude extract (Hh.Cr), (**G**) Ethyl acetate fraction (Ea.Hh), (**H**) Aqueous fraction (Ea.Aq), and (**I**) verapamil on K^+^ Induced Contraction on rabbit jejunum preparations.

**Figure 3 plants-11-00078-f003:**
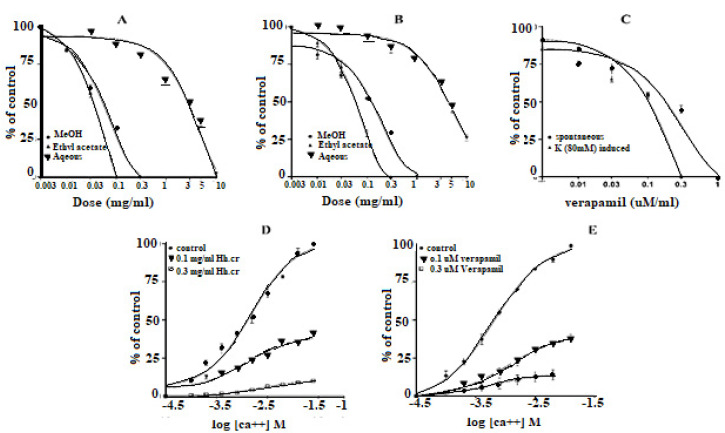
Effect of crude extract (Hh.Cr) Ethyl acetate fraction (Ea.Hh) and aqueous fraction (Ea.Aq) on (**A**) spontaneous contraction and (**B**) K^+^ (80 mM)-induce Contraction on rabbit jejunum preparations. (**C**) Effect of verapamil on spontaneous and K^+^ induced contraction on rabbit jejunum preparations. Dose–response curves of Ca^++^ in the presence and absence of (**D**) Hh.Cr (**E**) verapamil in the isolated rabbit jejunum preparations. Values are expressed as mean ± SEM.

**Figure 4 plants-11-00078-f004:**
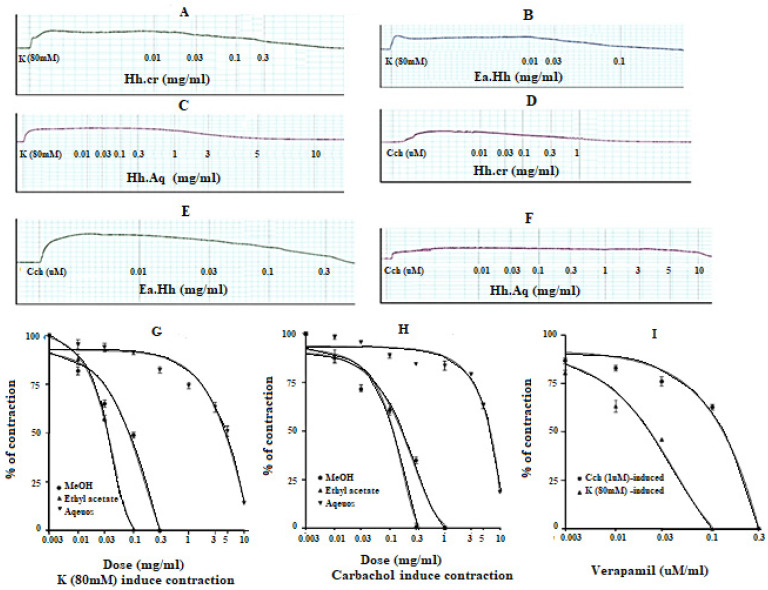
Effect of (**A**) crude extract (Hh.Cr) (**B**) Ethyl acetate fraction (Ea.Hh) (**C**) Aqueous fraction (Ea.Aq) on K^+^ Induce Contraction and Effect of (**D**) crude extract (Hh.Cr) (**E**) Ethyl acetate fraction (Ea.Hh) (**F**) Aqueous fraction (Ea.Aq) on CCh-induced contraction on rabbit tracheal preparations. Effect of crude extract (Hh.Cr) ethyl acetate fraction (Ea.Hh) and aqueous fraction (Ea.Aq) of on (**G**) K^+^ (80 mM) Induce Contraction and (**H**) CCh-induced contraction on tracheal preparations. (**I**) Effect of verapamil on CCh1 µM and K^+^ induced contraction on rabbit tracheal preparations. Values are expressed as mean ± SEM.

**Figure 5 plants-11-00078-f005:**
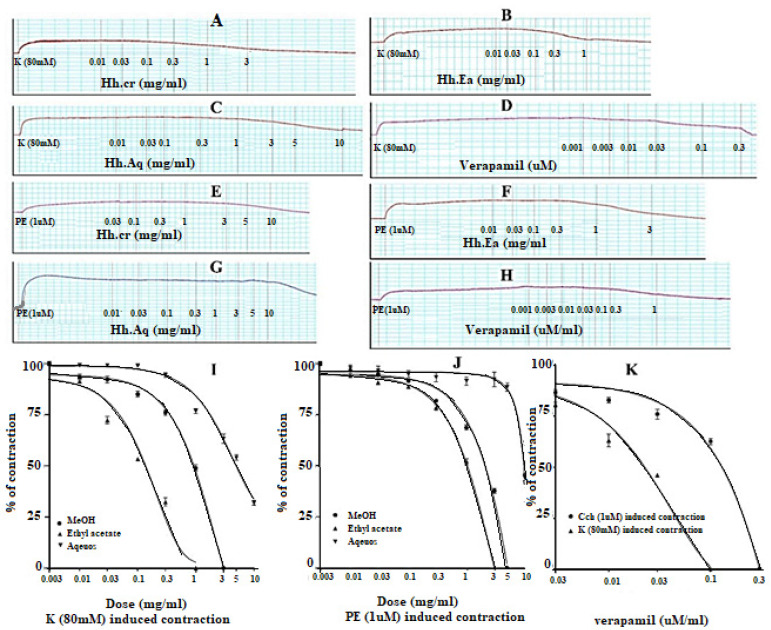
Effect of (**A**) crude extract (Hh.Cr), (**B**) Ethyl acetate fraction (Ea.Hh), (**C**) Aqueous fraction (Ea.Aq), and (**D**) verapamil on K^+^ induced contraction and effect of (**E**) crude extract (Hh.Cr), (**F**) Ethyl acetate fraction (Ea.Hh), (**G**) Aqueous fraction (Ea.Aq), and (**H**) verapamil on PE 1 µM Induced Contraction on rabbit aorta preparations. Effect of crude extract (Hh.Cr) Ethyl acetate fraction (Ea.Hh) and Aqueous fraction (Ea.Aq) on (**I**) K^+^ (80 mM) Induced Contraction and (**J**) PE 1 µM Induced Contraction on aortic jejunum preparations. (**K**) Effect of verapamil on PE 1 µM and K^+^ Induced Contraction on rabbit tracheal preparations. Values are expressed as mean ± SEM.

**Figure 6 plants-11-00078-f006:**
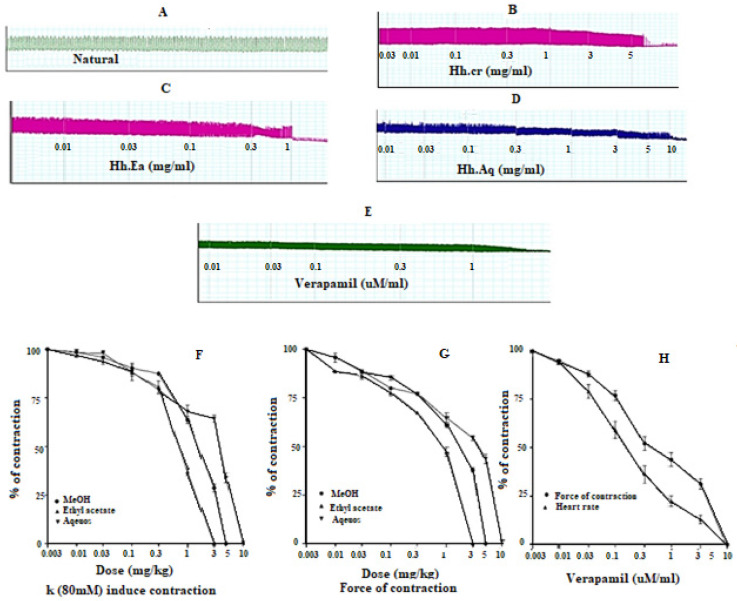
(**A**) Spontaneous contraction. Effect of (**B**) crude extract (Hh.Cr), (**C**) Ethyl acetate fraction (Ea.Hh), and (**D**) Aqueous fraction (Ea.Aq) on spontaneous contraction rabbit paired atrium preparations. (**E**) Effect of verapamil on spontaneous contraction rabbit paired atrium preparations. Effect of crude extract (Hh.Cr) Ethyl acetate fraction (Ea.Hh) and Aqueous fraction (Ea.Aq) on (**F**) K^+^(80mM)-induced contraction. (**G**) force of contraction. (**H**) Effect of verapamil on the force of contraction and heart rate on rabbit atrium preparations. Values are expressed as mean ± SEM.

**Figure 7 plants-11-00078-f007:**
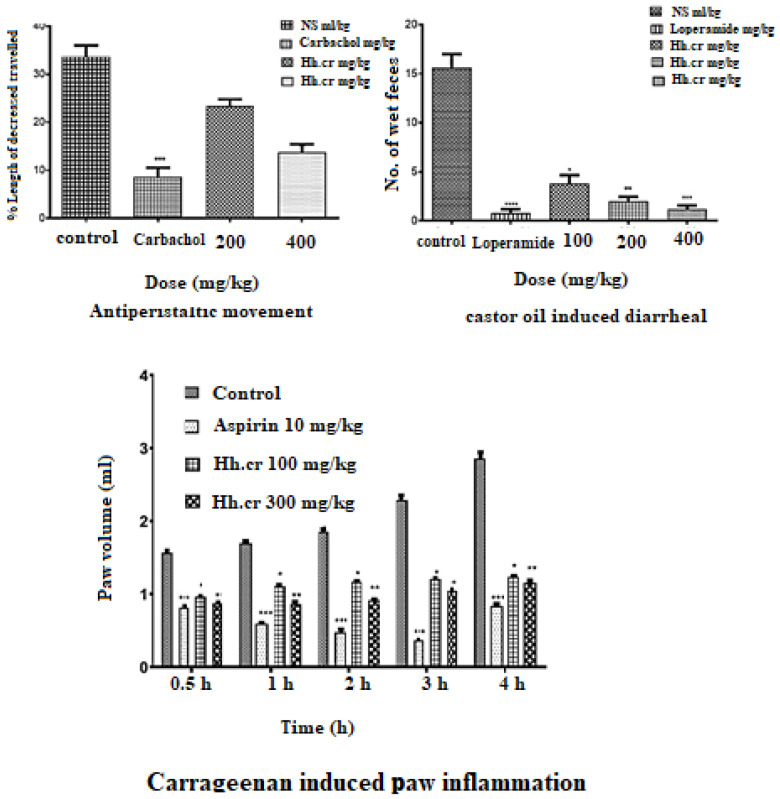
GI Charcoal meal transit (antiperistalsis) activity, castor oil-induced diarrhea activity, and carrageenan induce inflammation. Values are expressed as Mean ± SEM, and data was analyzed One way ANOVA or Two way ANOVA; * *p* < 0.05, ** *p* < 0.005, *** *p* < 0.0005 and **** *p* < 0.0001.

**Figure 8 plants-11-00078-f008:**
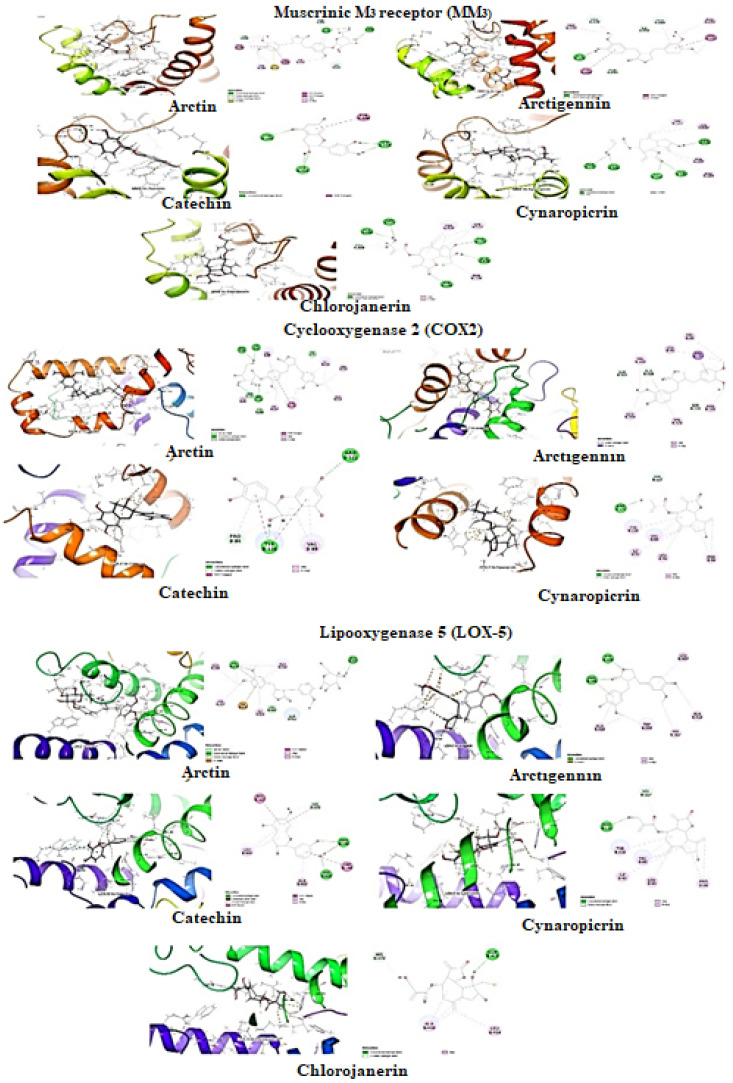
Molecular docking of selected compounds against muscarinic receptor, cyclooxygenase-2, and lipoxygenase 5.

**Table 1 plants-11-00078-t001:** Phenolic and flavonoid compounds of Hh.Cr.

Sr.No.	Compound	Retention Time (min)	Concentration (µg/g)
1.	Gallic Acid	2.7	184.98
2.	Catechin	3.3	160.37
3.	HB acid	6.8	22.80
4.	Vanilic acid	8.1	9.08

**Table 2 plants-11-00078-t002:** Binding energies (kcal/mol) of compounds with Muscarinic-3 (MM3, PDB ID: 4U14), Cyclooxygenase-2 (COX-2, PDB ID:5IKQ) and lipoxygenase 5 (LOX-5, PDB ID: 6N2W) receptors calculated by Prime MMGBSA.

Name(PubChem ID)	Docking Score	∆G_Binding_	Log K_i_ (µMolar)	∆G_Coulomb_	∆G_Covalent_	∆G_Hbond_	∆G_Lipophilic_	∆G_Solv GB_	∆G_vdW_	Residue-Ligand Interactions with Distance (Å)	
										Hydrogen Bonds	Hydrophobic Bonds
**Arctiin** **(100528)**	**−11.63**	**−60.79**	**−23.17**	**−16.29**	**10.56**	**−1.22**	**−42.47**	**47.40**	−55.32	Asn513 (2.46), Leu225 (1.77), **C-H Bond:** Thr231 (2.58), Tyr529 (2.99), **π-Donor Hydrogen Bond:** Trp525 (2.71), Trp525 (2.66)	**π-Sulfur Bond:** Cys532 (5.42), **π-π Stacked Bond:** Trp503 (4.56), **π-π T shaped Bond:** Tyr148 (5.21), **Alkyl Bond:** Ile222 (4.69), Leu225 (4.41), Cys532 (3.20), **π-Alkyl Bond:** Tyr148 (3.64), Tyr506 (3.99), Tyr529 (4.91), Tyr529 (3.98), Leu225 (4.98)
Arctigenin (64981)	−9.72	−46.56	−16.99	−16.92	6.57	−0.68	−30.61	31.10	−34.45	Ala238 (1.64) **C-H Bond:** Thr234 (3.03), Ile222 (2.77), Leu225 (2.77), Leu225 (2.65), Tyr148 (2.40)	**π-π T shaped Bond:** Trp503 (5.58), Trp525 (5.37), **π-Alkyl Bond:** Tyr148 (4.40), Trp199 (4.19), Trp199 (4.18), Phe221 (4.72), Trp525 (3.99), Trp525 (4.05), Leu225 (5.49), Ala238 (4.30)
Catechin (9064)	−7.59	−52.22	−19.45	−29.22	2.89	−2.92	−13.93	27.23	−34.12	Tyr148 (2.05), Ile222 (2.39), Ile222 (3.03), Ser226 (1.92), Ser226 (1.80)	**π-π T shaped Bond:** Tyr506 (5.74)
Chlorojanerin (182408)	−7.13	−43.68	−15.74	−20.79	2.92	−1.98	−19.31	33.51	−38.02	Tyr127 (1.87), Tyr148 (2.30), Asn513 (3.02) Asn513 (3.03), Asn526 (2.03), Ser226 (2.60), **C-H Bond:** Leu225 (2.63), Leu225 (2.55), Ser226 (2.55)	**Alkyl Bond:** Lys522 (5.21), Lys522 (5.47), **π-Alkyl Bond:** Phe124 (4.85), Trp525 (4.93), Trp525 (5.15), Trp525 (3.59), Trp525 (4.43)
Cynaropicrin (119093)	−6.76	−48.69	−17.92	−19.54	1.54	−1.63	−18.37	26.11	−36.81	Tyr148 (3.00), Ile222 (2.49), Asn526 (1.77), Leu225 (1.92), Thr231 (2.64)	**Alkyl Bond:** Lys522 (5.30), **π-Alkyl Bond:** Phe124 (5.48), Tyr127 (4.81), Trp525 (3.49), Trp525 (4.36)
**Cyclooxygenase-2 (COX-2, PDB ID:5IKQ)**											
Arctiin (100528)	−8.49	−41.01	−14.58	−21.01	10.60	−2.56	−23.77	33.40	−36.96	Lys83 (1.88), Ser12 (3.00), Ser120 (1.73), Pro84 (1.67), **C-H Bond:** Ser120 (2.66)	**π-π T shaped Bond:** Tyr11 (5.32), **Alkyl Bond:** Ala112 (3.50), Val89 (4.95), Leu93 (4.90), Val117 (4.37), Leu109 (4.79), Ile113 (5.11), **π-Alkyl Bond:** Tyr116 (4.01), Val89 (3.76), Le113 (4.71)
Arctigenin (64981)	−7.37	−27.91	−8.89	−5.55	18.02	0.00	−31.09	26.83	−35.44	**C-H Bond:** Ala528 (2.91), Ser120 (2.62), Ser531 (2.79)	**π-σ Bond:** Val117 (2.48), **Alkyl Bond:** Arg121 (4.83), Val350 (4.80), Leu353 (5.46), Val89 (4.59), Leu93 (5.01), **π-Alkyl Bond:** Val350 (5.15), Leu353 (4.99), Val524 (4.56) Ala528 (4.24)
Cynaropicrin (119093)	−4.28	−35.97	−12.39	−11.96	3.10	−1.10	−18.58	17.86	−25.30	Arg121(1.81), Arg121 (2.46), **C-H Bond:** Val117 (2.51)	**Alkyl Bond:** Pro84 (5.14), Val89 (5.04), Val89 (4.09), Pro84 (4.81), Val89 (4.45), Ile92 (5.00), Leu93 (3.76), **π-Alkyl Bond:** Tyr116 (5.22)
Catechin (9064)	−2.84	−11.87	−1.93	2.05	4.15	−0.61	−11.82	15.82	−18.71	Arg121 (2.76), Tyr116 (2.79), **C-H Bond:** Pro84 (2.55), Tyr116 (2.29)	**π-π T shaped Bond:** Tyr116 (5.65), Tyr116 (4.80), **Alkyl Bond:** Val89 (4.00), **π-Alkyl Bond:** Tyr116 (5.27), Val89 (4.80), Pro84 (5.20)
**Lipoxygenase 5 (LOX-5, PDB ID: 6N2W)**											
Arctiin (100528)	−5.76	−30.76	−10.13	−14.19	6.04	−2.11	−16.17	44.56	−46.53	His372 (2.55), Glu417 (1.89), **C-H Bond:** Glu417 (3.00), Gln413 (2.62)	**Electrostatic π-Anion Bond:** Ile673 (4.46) **π-π Stacked Bond:** His372 (4.40), **Alkyl Bond:** Ala410 (3.62), Leu368 (4.59), Leu368 (4.40), **π-Alkyl Bond:** His367 (3.52), His372 (4.83), His372 (3.83), Ile406 (5.49), Ala410 (4.09)
Catechin (9064)	−4.95	−30.81	−10.15	−22.76	5.66	−2.38	−15.10	39.78	−32.68	Arg596 (2.34), His600 (1.80), **π-Donor Hydrogen Bond:** His372 (3.10)	**π-π Stacked Bond:** His367 (4.84), **π-π T shaped Bond:** His372 (5.54), Trp599 (5.02), **Alkyl Bond:** Leu607 (5.03), **π-Alkyl Bond:** Leu607 (5.21), Ala603 (4.88)
Arctigenin (64981)	−4.84	−42.94	−15.42	−28.08	3.66	−3.06	−18.99	34.43	−29.61	Arg596 (2.57), Arg596 (1.88), His600 (1.82)	**Electrostatic π-Cation Bond:** Arg596 (3.17), **Alkyl Bond:** Ala410 (3.75), Ala426 (3.63), **π-Alkyl Bond:** His367 (4.32), Trp599 (4.00), Leu607 (5.10), Ala426 (3.91)
Cynaropicrin (119093)	−3.45	−14.54	−3.09	−13.54	2.95	−0.81	−17.02	46.96	−33.09	His367 (2.76), Ile673 (1.84), **C-H Bond:** Ala410 (2.98)	**Alkyl Bond:** Ala603 (4.92), Ala603 (3.87), Leu607, (5.09), Leu607 (4.33)
Chlorojanerin (182408)	−3.30	−32.60	−10.93	−3.30	0.38	−0.62	−13.65	19.69	−35.10	Thr427 (2.78), Arg596 (1.77), His600 (2.15), **C-H Bond:** His367 (2.97), His600 (2.56), Pro569 (2.73)	**π-π T shaped Bond:** Trp599 (5.64), **Alkyl Bond:** Ala603 (3.39), Val604 (4.44), **π-Alkyl Bond:** His360 (5.21), His432 (4.65), Trp599 (4.91) His600 (4.18)

∆G_Binding_: Binding free energy, Log Ki: Logarithmic of Inhibition Constant (K_i_), ∆G_Coulomb_: Coulomb binding energy, ∆G_Covalent_: Covalent binding energy ∆G_Hbond_: Hydrogen bonding energy, ∆G_Lipophilic_: Lipophilic binding energy, ∆G_Solv GB_: Generalized born electrostatic solvation energy ∆G_vdW_: Van der Waals forces energy, and C-H Bond: Carbon–Hydrogen Bond. These all contribute to Binding free energy (∆G_Binding_).

## Data Availability

Not applicable.

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
