# Peer review of "Antidiarrheal and Cardio-Depressant Effects of Himalaiella heteromalla (D.Don) Raab-Straube: In Vitro, In Vivo, and In Silico Studies"

_plants, 2021, doi:10.3390/plants11010078_

Round 1
Reviewer 1 Report
The manuscript "Antidiarrheal and cardio-depressant effects of Himalaiella heteromalla (D.Don) Raab-Straube: in vitro, in vivo and in silico studies” is very interesting and the "in vivo" experiments demonstrate the interesting biological properties of Himalaiella heteromalla (D.Don) Raab-Straube and its extract. However, the "in vivo" and "in vitro" experiments are complex in their programming and therefore I believe that the doses and times chosen have been carefully planned. How were the doses of 100-400 mg / kg determined? Silico studies have highlighted the possible biological role of some molecules present in the extract. Do you believe that their presence in the extract can play a leading role in the biological activity manifested by the extract, or should we speak more properly of a synergistic effect operated by several molecules? Furthermore, it would be useful to know whether the biochemical parameters of the blood, in the presence of the inducers and of the extract, were determined during the "in vivo" experimentation. In fact, I might think, for example, that the extract administered to animals may also limit the effects of aspirin on the number of platelets.
Author Response
The Editor
Plants
Subject: Submission of revised manuscript ID: plants-1521682[6159]
Dear Sir,
It is stated that I want to submit revised manuscript of article entitled, “Antidiarrheal and cardio-depressant effects of Himalaiella heteromalla (D.Don) Raab-Straube: in vitro, in vivo and in silico studies”. I am highly thankful to referees whose comments helped in improving this manuscript. Below is response to referee comments:
Reviewer 1
Reviewer’s comments :The manuscript "Antidiarrheal and cardio-depressant effects of Himalaiella heteromalla (D.Don) Raab-Straube: in vitro, in vivo and in silico studies” is very interesting and the "in vivo" experiments demonstrate the interesting biological properties of Himalaiella heteromalla (D.Don) Raab-Straube and its extract. However, the "in vivo" and "in vitro" experiments are complex in their programming and therefore I believe that the doses and times chosen have been carefully planned.
Author’s response: The authors are grateful for the insightful comments of the reviewer on the scientific merit of our work.
For the in vitro and in vivo experiments, we used established protocols as previously published internationally. We had planned the doses and times by following standardized international protocols. (Saqib and Janbaz, 2021, 2016; Wahid et al., 2021; Janbaz et al., 2015; Mehmood et al., 2014; Gilani et al., 2008)
Reference
Gilani, A.H., Mehmood, M.H., Janbaz, K.H., Khan, A., Saeed, S.A., 2008. Ethnopharmacological studies on antispasmodic and antiplatelet activities of Ficus carica. J. Ethnopharmacol. 119, 1–5. https://doi.org/10.1016/j.jep.2008.05.040
Janbaz, K.H., Zaeem Ahsan, M., Saqib, F., Imran, I., Zia-Ul-Haq, M., Abid Rashid, M., Jaafar, H.Z.E., Moga, M., 2015. Scientific basis for use of Pyrus pashia Buch.-Ham. ex D. Don. fruit in gastrointestinal, respiratory and cardiovascular ailments. PLoS One 10, e0118605. https://doi.org/10.1371/journal.pone.0118605
Mehmood, M.H., Anila, N., Begum, S., Syed, S.A., Siddiqui, B.S., Gilani, A.H., 2014. Pharmacological basis for the medicinal use of Carissa carandas in constipation and diarrhea. J. Ethnopharmacol. 153, 359–367. https://doi.org/10.1016/j.jep.2014.02.024
Saqib, F., Janbaz, K.H., 2021. Ethnopharmacological basis for folkloric claims of Anagallis arvensis Linn. (Scarlet Pimpernel) as prokinetic, spasmolytic and hypotensive in province of Punjab, Pakistan. J. Ethnopharmacol. 267, 113634. https://doi.org/10.1016/j.jep.2020.113634
Saqib, F., Janbaz, K.H., 2016. Rationalizing ethnopharmacological uses of Alternanthera sessilis: A folk medicinal plant of Pakistan to manage diarrhea, asthma and hypertension. J. Ethnopharmacol. 182, 110–121. https://doi.org/10.1016/j.jep.2016.02.017
Wahid, M., Saqib, F., Ahmedah, H.T., Gavris, C.M., Feo, V. De, Hogea, M., Moga, M., Chicea, R., 2021. Cucumis sativus L. Seeds Ameliorate Muscular Spasm-Induced Gastrointestinal and Respiratory Disorders by Simultaneously Inhibiting Calcium Mediated Signaling Pathway. Pharmaceuticals 14, 1197. https://doi.org/10.3390/PH14111197
Reviewer’s comments: How were the doses of 100-400 mg / kg determined?
Authors response: Doses were selected by repeated trials on animals. In vivo doses are calculated on the basis of body weight of the rat. Doses which produced effective response were found to be 100-400mg/kg. ( Muqeet et al., 2021; Amira et al.,2008; Gilani et al.,2008).
References :
Amira, S.; Rotondo, A.; Mulè, F. Relaxant effects of flavonoids on the mouse isolated stomach: Structure-activity relationships. Eur. J. Pharmacol. 2008, 599, 126–130, doi:10.1016/j.ejphar.2008.09.021.
Gilani, A.H.; Khan, A.U.; Raoof, M.; Ghayur, M.N.; Siddiqui, B.S.; Vohra, W.; Begum, S. Gastrointestinal, selective airways and urinary bladder relaxant effects of Hyoscyamus niger are mediated through dual blockade of muscarinic receptors and Ca2+ channels. Fundam. Clin. Pharmacol. 2008, 22, 87–99, doi:10.1111/j.1472-8206.2007.00561.x.
Wahid, M., Saqib, F., Ahmedah, H.T., Gavris, C.M., Feo, V. De, Hogea, M., Moga, M., Chicea, R., 2021. Cucumis sativus L. Seeds Ameliorate Muscular Spasm-Induced Gastrointestinal and Respiratory Disorders by Simultaneously Inhibiting Calcium Mediated Signaling Pathway. Pharmaceuticals 14, 1197. https://doi.org/10.3390/PH14111197.
Reviewer’s comments : Silico studies have highlighted the possible biological role of some molecules present in the extract. Do you believe that their presence in the extract can play a leading role in the biological activity manifested by the extract, or should we speak more properly of a synergistic effect operated by several molecules?
Authors response:
*Yes, presence of biological molecules in the extract can play a leading role in the biological activity manifested by the extract.
*In the current work, we have quantified the Gallic Acid, Catechin, HB acid and Vanilic acid in crude extract through HPLC. These compounds have been reported to possess different biological activities. For example Gallic acid is reported to possess anti-inflammatory and antioxidant activity (Karimi-Khouzani et al., 2017), anti-allergic activity of gallic acid. (Liu et al., 2013) Catechin attenuates TNF-α induced inflammatory response(Cheng et al., 2019). Whereas catechin also reported to possess antispasmodic, bronchodilator, and vasodilator activities of catechin. (Ghayur et al., 2007).
*Our crude extract found to have multiple constituents which may have synergistic action.In scientific literature, it was previously reported that quercetin and rutin have synergistic activities, similary apigenin, genistein, quercetin, rutin, naringenin, caffeic acid, and catechin had synergistic effects (Amira et al., 2008) , (Lanuzza et al., 2017), (Ghayur et al., 2007), (Ajay et al., 2003; Gilani et al., 2006)
Reference:
Ajay, M., Gilani, A.U.H., Mustafa, M.R., 2003. Effects of flavonoids on vascular smooth muscle of the isolated rat thoracic aorta. Life Sci. 74, 603–612. https://doi.org/10.1016/j.lfs.2003.06.039
Amira, S., Rotondo, A., Mulè, F., 2008. Relaxant effects of flavonoids on the mouse isolated stomach: Structure-activity relationships. Eur. J. Pharmacol. 599, 126–130. https://doi.org/10.1016/j.ejphar.2008.09.021
Cheng, A.-W., Tan, X., Sun, J.-Y., Gu, C.-M., Liu, C., Guo, X., 2019. Catechin attenuates TNF-α induced inflammatory response via AMPK-SIRT1 pathway in 3T3-L1 adipocytes. PLoS One 14, e0217090. https://doi.org/10.1371/journal.pone.0217090
Ghayur, M.N., Khan, H., Gilani, A.H., 2007. Antispasmodic, bronchodilator and vasodilator activities of (+)-catechin, a naturally occurring flavonoid. Arch. Pharm. Res. 30, 970–975. https://doi.org/10.1007/BF02993965
Gilani, A.H., Khan, A.U., Ghayur, M.N., Ali, S.F., Herzig, J.W., 2006. Antispasmodic effects of Rooibos tea (Aspalathus linearis) is mediated predominantly through K+-channel activation. Basic Clin. Pharmacol. Toxicol. 99, 365–373. https://doi.org/10.1111/j.1742-7843.2006.pto_507.x
Karimi-Khouzani, O., Heidarian, E., Amini, S.A., 2017. Anti-inflammatory and ameliorative effects of gallic acid on fluoxetine-induced oxidative stress and liver damage in rats. Pharmacol. Reports 69, 830–835. https://doi.org/10.1016/j.pharep.2017.03.011
Liu, K., Hu, S., Chan, B., Wat, E., Lau, C., Hon, K., Fung, K., Leung, P., Hui, P., Lam, C., Wong, C., 2013. Anti-Inflammatory and Anti-Allergic Activities of Pentaherb Formula, Moutan Cortex (Danpi) and Gallic Acid. Molecules 18, 2483–2500. https://doi.org/10.3390/molecules18032483
Lanuzza, F., Occhiuto, F., Monforte, M.T., Tripodo, M.M., D’Angelo, V., Galati, E.M., 2017. Antioxidant phytochemicals of Opuntia ficus-indica (L.) Mill. cladodes with potential antispasmodic activity. Pharmacogn. Mag. 13, S424–S429. https://doi.org/10.4103/pm.pm_495_16
Reviewer’s comments: Furthermore, it would be useful to know whether the biochemical parameters of the blood, in the presence of the inducers and of the extract, were determined during the "in vivo" experimentation. In fact, I might think, for example, that the extract administered to animals may also limit the effects of aspirin on the number of platelets.
Author’s response:
*The anti-inflammatory activity was performed on 25 rats divided into 5 groups( each group containing 5 rats) i.e.one group as control (0.9% normal saline), 2nd group as standard drug (aspirin 10mg/kg), and 3rd,4th,5th group receiving only crude extract H. heteromalla doses (100, 200 and 400 mg/kg).
* As Aspirin is the standard drug given only to standard group. Standard group is different from treated group.The treated group only received crude extract. The percentage of edema inhibition was calculated in each group separately and compared with standard drug.So biochemical parameters of blood need not to be estimate because anti-inflammatory activity of Aspirin is measured in different group of animals from treated group.( Orhan et al., 2007)
Refrence
Orhan, D.; Hartevioǧlu, A.; Küpeli, E.; Yesilada, E. In vivo anti-inflammatory and antinociceptive activity of the crude extract and fractions from Rosa canina L. fruits. J. Ethnopharmacol. 2007, 112, 394–400, doi:10.1016/j.jep.2007.03.029
Reviewer 2
Reviewer’s comments: Himalaiella heteromalla (D.Don) Raab-Straube is a commonly used remedy against various diseases. Crude extract and fractions of H. heteromalla were investigated for gastrointestinal, bronchodilator, cardiovascular, and anti-inflammatory activities. H. heteromalla crude extract (Hh.Cr) relaxed spontaneous contractions and K+ (80mM)-induced contraction in jejunum tissue dose-dependently. Application of Hh.Cr on aortic preparations exhibited vasorelaxant activity through angiotensin and α-adrenergic receptors blockage. It also showed the cardio suppressant effect with negative chronotropic and inotropic response in paired atrium preparation.
Compared with existing reports, the present topic is very interesting, the method used in the research is novel and reliable, and the conclusion was supported by the results resulting in the experiments. This study has significantly contributed to the research of plants, and especially for the pharmacological research of Himalaiella heteromalla. This manuscript can be accepted after minor revision. The reviewer would like to review the revision again.
Authors response: Thank you very much for your appreciation of our scientific research work. We are highly obliged for this gratitude.
Reviewer’s comments: Apart from grammar editing, several suggestions were given on this manuscript.
Authors response: : The entire manuscript has been rewritten and extensively edited for English errors. GRAMMARLY Premium Service was used to make manuscript uptomark. (All corrections are shown in red font).
Reviewer’s comments: Determination of water content in the Himalaiella heteromalla samples will help the test results to be more accurate. Although no additional experiments need to be supplemented, the manuscript need discuss is there any potential approaches to determine water content. This report can be referenced: doi: 10.25165/j.ijabe.20191206.4914.
Authors response: We have now discussed the determination of water content as per the reviewer suggestion please refer line 253-255
Reviewer’s comments: Although no additional experiments need to be supplemented, the manuscript should discuss is there any new and potential approaches to pre-treat and extract the active components from the Himalaiella heteromalla samples as much as possible, such as presoaking, liquid ammonia pretreatment, and others. These reports can be referenced: doi.org/10.1016/j.biombioe.2017.01.001; doi.org/10.1021/acs.energyfuels.8b00951; doi.org/10.1016/j.rser.2020.110444.
Authors response: By refreing given link (,doi.org/10.1016/j.biombioe.2017.01.001; doi.org/10.1021/acs.energyfuels.8b00951; doi.org/10.1016/j.rser.2020.110444), We have now discussed the to pretreatment of extract as per the reviewer suggestion please refer line 391-393
Reviewer’s comments In this study, the plant material was ground, and then the powder was macerated in methanol aqueous (70:30) for maceration extraction. Although no additional experiments need to be supplemented, the manuscript need discuss is there any potential different drying pretreatment methods for the Himalaiella heteromalla samples. These reports can be referenced: doi.org/10.1016/j.biortech.2020.123130; doi.org/10.1016/j.fuel.2020.117936.
Authors response: By refreing given link (doi.org/10.1016/j.biortech.2020.123130; doi.org/10.1016/j.fuel.2020.117936),We have discussed the pretreatment of drying of extract as per the reviewer suggestion please refer line 393-396
The manuscript can also discuss different methods of removing metals. This report can be referenced: doi.org/10.1016/j.fuel.2019.116347.
Authors response: By refreing given link (doi.org/10.1016/j.fuel.2019.116347), We have discussed the different methods of removing metals from extract as per the reviewer suggestion please refer line 393-396
Reviewer’s comments: The resolution of the figures should be improved, and the format of the reference list needs to be checked again.
Authors response:
By following the instructions of reviewer,
- We have improved the resolution of figures
- We have also rechecked and formatted the reference list
PS.We have added author Muhammad Riaz which we forgot to add previously.
We are again thankful for referee comments.I amavailable if there is anything more required.
-----
Regards

Reviewer 2 Report
Himalaiella heteromalla (D.Don) Raab-Straube is a commonly used remedy against various diseases. Crude extract and fractions of H. heteromalla were investigated for gastrointestinal, bronchodilator, cardiovascular, and anti-inflammatory activities. H. heteromalla crude extract (Hh.Cr) relaxed spontaneous contractions and K+ (80mM)-induced contraction in jejunum tissue dose-dependently. Application of Hh.Cr on aortic preparations exhibited vasorelaxant activity through angiotensin and α-adrenergic receptors blockage. It also showed the cardio suppressant effect with negative chronotropic and inotropic response in paired atrium preparation.
Compared with existing reports, the present topic is very interesting, the method used in the research is novel and reliable, and the conclusion was supported by the results resulting in the experiments. This study has significantly contributed to the research of plants, and especially for the pharmacological research of Himalaiella heteromalla. This manuscript can be accepted after minor revision. The reviewer would like to review the revision again. Apart from grammar editing, several suggestions were given on this manuscript.
1, Determination of water content in the Himalaiella heteromalla samples will help the test results to be more accurate. Although no additional experiments need to be supplemented, the manuscript need discuss is there any potential approaches to determine water content. This report can be referenced: doi: 10.25165/j.ijabe.20191206.4914.
2, Although no additional experiments need to be supplemented, the manuscript should discuss is there any new and potential approaches to pre-treat and extract the active components from the Himalaiella heteromalla samples as much as possible, such as presoaking, liquid ammonia pretreatment, and others. These reports can be referenced: doi.org/10.1016/j.biombioe.2017.01.001; doi.org/10.1021/acs.energyfuels.8b00951; doi.org/10.1016/j.rser.2020.110444.
3, In this study, the plant material was ground, and then the powder was macerated in methanol aqueous (70:30) for maceration extraction. Although no additional experiments need to be supplemented, the manuscript need discuss is there any potential different drying pretreatment methods for the Himalaiella heteromalla samples. These reports can be referenced: doi.org/10.1016/j.biortech.2020.123130; doi.org/10.1016/j.fuel.2020.117936. The manuscript can also discuss different methods of removing metals. This report can be referenced: doi.org/10.1016/j.fuel.2019.116347.
4, The resolution of the figures should be improved, and the format of the reference list needs to be checked again.
Author Response

(The authors gave the same response as above.)

Round 2
Reviewer 1 Report
I thank the authors for answering my questions and including the bibliography.